# Transfluthrin and Metofluthrin as Effective Repellents against Pyrethroid-Susceptible and Pyrethroid-Resistant *Aedes aegypti* (L.) (Diptera: Culicidae)

**DOI:** 10.3390/insects14090767

**Published:** 2023-09-14

**Authors:** Dae-Yun Kim, Jeffrey Hii, Theeraphap Chareonviriyaphap

**Affiliations:** 1Department of Entomology, Faculty of Agriculture, Kasetsart University, Bangkok 10900, Thailand; daeyun.k@ku.th; 2College of Public Health, Medical & Veterinary Sciences, James Cook University, Brisbane, QLD 4000, Australia; hiijk1@gmail.com

**Keywords:** *Aedes aegypti*, spatial repellent, high-throughput screening system, toxicity bioassay, transfluthrin, metofluthrin

## Abstract

**Simple Summary:**

*Aedes aegypti* (L.) is a major vector of dengue fever in tropical regions. To prevent their contact with human hosts, spatial repellents (SRs) like transfluthrin (TFT) and metofluthrin (MFT) have shown promise in delaying pesticide resistance and addressing gaps in outdoor transmission not covered by other interventions such as indoor residual spray or long-lasting insecticide-treated nets. In this study, we successfully identified optimal discriminating concentrations of TFT and MFT, using a high-throughput screening system toxicity bioassay (HITSS-TOX). These concentrations were effective against both susceptible and resistant *Ae. aegypti* mosquitoes. However, it was observed that TFT required a 4.7-fold higher concentration compared to MFT. Additionally, after 60 min of exposure, TFT caused a stronger knockdown (KD_60_) of mosquitoes but did not significantly increase the 24 h mortality rate compared to MFT. This means that more mosquitoes exposed to TFT were able to recover from KD_60_, unlike those exposed to MFT. To better understand the behavioral response of mosquitoes to these repellents, further research is required using the HITSS contact irritancy and spatial repellency assays. Such investigations could provide valuable insights into improving vector control strategies and combating the transmission of dengue fever and other mosquito-borne diseases.

**Abstract:**

*Aedes aegypti* is a major vector of dengue fever in tropical regions. Spatial repellents (SRs) have shown promise in delaying pesticide resistance. Methods for discriminating concentrations (DCs) are well established using various bioassay tools, while data for high-throughput screening system (HITSS) toxicity bioassay (TOX) are absent. In this study, we compared and optimized lethal (LCs) and sub-lethal concentrations (SLCs) of transfluthrin (TFT) and metofluthrin (MFT) on pyrethroid-susceptible (USDA) and pyrethroid-resistant (Pu-Teuy) *Ae. aegypti* (L.) strains, using the HITSS-TOX. Mean mortality (MT) was 100% at LC_99_ and DC, compared to LC_50_ (45.0 ± 3.7%) and LC_75_ (65.8 ± 7.0%) for the USDA strain. However, the resistant strain (Pu-Teuy) showed reduced susceptibility against TFT and a significantly lower MT at LC_50_ (12.5 ± 4.4%; t = 5.665, df = 10, *p* < 0.001), LC_75_ (9.2 ± 3.5%; t = 4.844, df = 10, *p* = 0.001), LC_99_ (55.0 ± 9.9%; t = 4.538, df = 5, *p* = 0.006), and DC (75.0 ± 5.2%; U = 3.0, *p* = 0.007). The DC of TFT (0.15222%) was 4.7-fold higher than for MFT (0.03242%) in USDA strain. The baseline DCs established are useful to better understand susceptibility and the efficacy of various repellents against field populations of *Ae. aegypti*.

## 1. Introduction

Dengue is the most rapidly spreading mosquito-borne viral disease, which is estimated to cause 390 million infections annually and approximately 20,000 deaths, of which 70% of the disease burden is reported from Asia [1]. In the absence of a fully effective vaccine against dengue fever, vector control strategies are currently deployed to prevent dengue transmission in situations where *Ae. aegypti* mosquitoes freely feed on human hosts both indoors and outdoors [2]. Synthetic pyrethroids such as deltamethrin, cypermethrin and permethrin are commonly used in Thai public health programs to control *Aedes aegypti* adults [3,4], and their continued use is associated with permethrin resistance throughout the country [5,6,7], and resistance to deltamethrin, bifenthrin, lambda-cyhalothrin and DDT [5,8,9,10].

Volatile pyrethroids spatial repellents (VPSRs) can prevent human–vector contact by disrupting hematophagous arthropods from entering the space occupied by human hosts and stimulating directed movement away from the source without physical contact [11]. As the effectiveness of VPSRs depends on the volume of indoor or outdoor space to be protected and the dose of the active ingredients (AIs), it is crucial to choose the appropriate AI based on its chemical and physical properties, the application platform, and local environmental conditions. Transfluthrin (TFT) and metofluthrin (MFT) are two of the main highly volatile pyrethroids (VPs) which have been applied in various formats, such as coils [12], kerosene mixture for lamps [13], electro-vaporizers [14], multi-layer paper strips [15], and hessian strips [16]. Like conventional insecticides, these volatile pyrethroids are subject to an increased risk of resistance [17,18]. Awareness and understanding of the level of resistance and resistance mechanisms in *Ae. aegypti* populations in Thailand are crucial for preventing instances like this.

As early detection of the pesticide resistance of field mosquito populations is essential for resistance management, various bioassays such as WHO bottles, CDC bottles, WHO tubes, and topical application bioassays are recommended by WHO [19]. For example, the field population of *Ae. aegypti* collected from Puerto Rico showed cross-resistance between permethrin and metofluthrin using CDC bottle assay, with a 6-fold higher susceptibility for metofluthrin relative to permethrin [20]. Another study using WHO tube bioassay confirmed that *Ae. aegypti* field population developed resistance towards permethrin (6% mortality) [21] and transfluthrin (56% mortality) [17]. Although both bioassays confirmed the relatively higher resistance to permethrin, the DCs were different for each bioassay due to the differential volume of the testing chamber. This anomaly emphasizes the importance of optimizing LCs and DCs of bioassay tools and minimizing variations in the testing apparatus [17].

The HITTS has the advantage of assessing not only TOX, but also behavioral responses, such as contact irritancy assay (CIA) and spatial repellency assay (SRA). Although the accuracy of the HITSS has been validated in the laboratory [21,22], semi-field [23], and field [24] settings, data is lacking for SRs, especially DCs for baseline susceptibility tests. Several bioassay methods have been used to measure the toxicity or spatial repellency of AIs used for SRs, such as HITSS-SRA [25], fumigant bioassay [26], and excito-repellency assay (ERA) [27], for non-contact, and WHO tube bioassay [17], and WHO bottle bioassay [18] for contact. However, among these assays, only a single device, HITSS, has the advantage of assessing three functions: TOX, CIA, and SRA. In addition, previous studies emphasized that obtaining optimal doses of the AIs between above the minimum repellency detection and below knockdown (KD) is important to observe and measure the mosquitoes’ behavioral responses within the HITSS device.

In this study, we aimed to test the dose-dependent responses of pyrethroid-susceptible and pyrethroid-resistant *Ae. aegypti* female mosquitoes to TFT and MFT using HITSS-TOX. First, the toxicity bioassay was conducted using five serial concentrations of TFT and MFT against susceptible *Ae. aegypti* to determine the LCs based on the 50%, 75%, and 99% mortality (MT) data. Second, DCs of each AI were obtained by doubling the 99% of LC to evaluate the accuracy of the HITSS-TOX by running susceptibility tests against the pyrethroid-susceptible and pyrethroid-resistant *Ae. aegypti*. Lastly, the contact toxicities of TFT and MFT were compared based on mosquito recovery rates (MRE) at 24 h post exposure which is recovered mosquitoes from an hour post-exposure KD (KD_60_) mosquito.

## 2. Materials and Methods

### 2.1. Mosquitoes

The *Aedes aegypti* laboratory strain (USDA) is so named as it was obtained (ca. 1996) from the United States Department of Agriculture (USDA), Gainesville, FL, USA. This insecticide-susceptible strain has been continuously maintained for over 20 years at the Department of Entomology, Faculty of Agriculture, Kasetsart University, Bangkok, Thailand under laboratory-controlled conditions. Field strain of *Ae. aegypti* larvae and pupae were collected in January 2019 from artificial containers near households in Pu-Tuey village, Kanchanaburi province, (14°170 N, 99°110 E), Western Thailand, and interbred with a newly collected field strain in the same area annually. Larvae and pupae were immediately transferred for initial rearing and F1 to F5 colonies were used at the Department of Entomology, Faculty of Agriculture, Kasetsart University, Bangkok, Thailand.

### 2.2. Mosquito Rearing

The immature stages were reared to adults under controlled conditions (25 ± 5 °C, 80 ± 10% relative humidity (% RH), and 12 h/12 h light/dark photoperiod). Adult mosquitoes were provided with cotton pads soaked with 10% sugar solution on the day of emergence and maintained in separate insectaries. Naturally inseminated females were permitted to feed on blood through an artificial membrane feeding system on day 3 post-emergence. Two days after blood-feeding, 10 cm diameter oviposition dishes containing moist, white filter paper were placed in the adult holding cages for oviposition. Eggs were air-dried at room temperature for 1–2 days to allow embryonic maturation before being immersed in clean water in individual rearing trays (30 cm [L] × 20 cm [W] × 5 cm [H]). In order to achieve consistent mosquito body sizes, approximately 200 larvae per tray were provided with a daily feeding regimen using a commercially available fish protein mixture (Optimum^TM^ Nishikigoi Carp Fish; Perfect Companion Group Co., Ltd.; Samutprakarn, Thailand). Adults were maintained in screen cages with a 10% sucrose solution before assays.

### 2.3. Chemicals

Technical grades of 97.9% transfluthrin (CAS 118712-89-3) and 96.4% metofluthrin (CAS 240494-70-6) obtained from Earth (Thailand) Company Limited (Bangkok, Thailand) were used to prepare stock solutions with a mixture of analytical grade acetone (Avantor Performance Materials, Inc.; Allentown, PA, USA) and silicone oil (Dow Corning^®^ 556 cosmetic grade; Dow Chemical Company and Corning, Inc.; Midland, MI, USA) with a density of 0.98 g/mL at 25 °C as the carrier at a ratio of 2.05:1.01 [17]. Serial dilutions comprising five concentrations were prepared for impregnation on filter papers: 0.00107%, 0.00213%, 0.00427%, 0.00853%, and 0.01706%.

### 2.4. Insecticide-Impregnated Filter Papers

To avoid crystallization of the pyrethroids on the rough surface, we applied silicone oil to filter papers [17,28]. Whatman No. 1 filter papers (Whatman International Ltd.; Banbury, UK) measuring 11 cm × 25 cm were impregnated with 10.10 mg of technical grade 97.9% TFT or 96.4% MFT with a 3.6 mg mixture of carrier oil per square centimeter using a micropipette [29]. Aliquots (3 mL each) of the five serial concentrations of each TFT and MFT solution were applied on separate filter papers and air-dried on metal pins on a holding rack for 1 h under laboratory conditions [30]. Discriminating concentrations (DCs) of TFT and MFT were prepared by doubling the 99% lethal concentration (LC_99_) [31] for susceptible *Ae. aegypti* (USDA). Untreated control papers were treated with acetone and silicone oil mixture (3.0 mL) only without AIs.

### 2.5. High-Throughput Screening System

The HITSS device consists of three interconnecting cylinders [30], which allow several testing options, such as toxicity (TOX) and two behavioral response assays: CIA and SRA [32]. The current study used TOX to obtain lethal concentrations of 50%, 75%, and 99%. The TOX comprised a single metal chamber with an end cap and a funnel section. Each metal chamber contained filter paper treated with either a single concentration of TFT or MFT for each treatment group. In the control group, the chamber contained filter paper treated with a solvent without any AI. Twenty nulliparous, mated, non-blood-fed female mosquitoes aged 3–5 days were provided with 10% sucrose solution on a moist cotton wick and starved for 12 h before testing for each replicate (provided with water only). These mosquitoes were transferred into the metal chamber using a mouth aspirator and forced into contact with the filter paper for 1 h, after which the number of KD_60_ mosquitoes was recorded. Test mosquitoes were kept in a holding cup at 25 ± 5 °C and 80 ± 5% RH with a 10% sugar-soaked cotton ball and checked for MT for 24 h. Six consecutive replicates were conducted for each of the five concentrations and the control simultaneously in order to reduce biological and environmental variations.

### 2.6. Data Analysis

The KD_60_ and 24 h MT of susceptible *Ae. aegypti* exposed to the five concentrations of both pyrethroids AIs were compared. Using the normality test result that did not assume equal variance (Shapiro–Wilk, *p* < 0.05), multiple comparisons were conducted for the non-parametric datasets using the Kruskal–Wallis H test (*p* < 0.05). Significance among the concentrations was tested using the comparison of mean rank with minimum and maximum ranges along with pairwise comparisons. Subsequently, the LCs of TFT and MFT at 50%, 75%, and 99% were obtained by running probit analysis on MT (number of female mosquitoes that had died at 24 h post exposure) at the five concentrations. To generate DCs, the MT data were analyzed using Pearson’s chi-square goodness-of-fit test based on comparisons of observed and expected distributions. In addition, 95% of fiducial limits were calculated from the baseline data using maximum likelihood estimates of parameters and log-probit regression analysis. As mentioned above, a doubling of the derived LC_99_ value was used as the final DC according to standard procedures [31]. Due to skewed toxicity data and to meet the assumptions of the statistical model, percentage KD_60_ and MT values were transformed to the natural logarithm of n as ln(n + 1). If the log-transformed data still did not produce an equal variance, further analysis was conducted on the original non-parametric data sets based on the Kruskal–Wallis H test for multiple comparisons or the Mann–Whitney *U* test for a 2-sample comparison. For normally distributed raw data, one-way ANOVA with Tukey’s honestly significant difference test (or Dunnett T3) and Student’s t-test was applied at *p* = 0.05 for multiple and 2-sample comparisons, respectively. Percentages of mean ± standard error (SE) of untransformed data were reported in the tables and figures. All statistical analyses were performed using the SPSS version 29 software (IBM; Armonk, NY, USA).

## 3. Results

### 3.1. Toxicity Assay to Generate Lethal Concentrations

The range of KD_60_ values between the lowest concentration of TFT (0.00107%, equivalent to 0.00359 mg/mL) TFT and the highest concentration (0.01706%, equivalent to 0.05719 mg/mL) was from 1.7 ± 1.1% to 89.2 ± 6.2%, respectively. These values exhibited significant differences. A similar trend was observed with MFT as KD_60_ significantly increased from 6.7 ± 2.5% at 0.00107% MFT to 99.2 ± 0.8% at 0.01706% MFT. At the highest concentration (0.01706%), the 24 h MT of MFT remained significantly higher compared to TFT (U = 0.0, *p* = 0.003) but KD_60_ was not significantly different (*p* = 0.072). Furthermore, neither of the AIs exhibited significant KD_60_ and MT values within the low concentration range of 0.00107% to 0.00427%, as indicated in Table 1. While the control group exhibited no KD_60_, and MT, the filter paper infused with SR had no discernible impact on the remaining mosquitoes.

The difference between KD_60_ and 24 h MT is referred to as the mosquito recovery rate (MRE), which is estimated by deducting MT% from KD% at each concentration for both VPs (Table 2). No significantly different MRE rates were observed between the two AIs except for the highest concentration (0.01706%) that was significantly different for MRE% (TFT: 20.0 ± 7.1%, MFT: 0.0 ± 0.0% (U = 6.0, *p* = 0.021), as shown in Table 2.

The MRE rates were directly proportional to the TFT concentrations (y = 1.2453 + 1.222.762x, R^2^ = 0.83; R-square determines coefficient, where a value of 1.0 indicates a perfect correlation) while the recovery rates plateaued at all MFT concentrations (y = 1.0478 + 71.3329x, R^2^ = 0.01), as shown in Figure 1. The HITSS assay was used to estimate the concentrations required to achieve 25% recovery by maintaining the exposure time at 60 min and varying the TFT or MFT concentrations. For TFT this occurred using 0.02046% (or 0.06859 mg/mL) which was 6.8-fold higher than for MTF (0.00299% or 0.01002 mg/mL) (Figure 1).

The LC_50_, LC_75_, and LC_99_ values were significantly different between TFT and MFT (LC_50_: 0.01040% vs. 0.00307%, LC_75_: 0.01852% vs. 0.00497%, and LC_99_: 0.7611% vs. 0.01621%), respectively. The DC of TFT (0.15222%) was about five-fold higher than for MFT (0.03242%), as shown in Table 3.

### 3.2. Toxicity Assay for Susceptibility Test Using Discriminating Concentrations

Probit analysis of the LCs and DCs showed a linear dose response for both USDA and Pu-Teuy *Ae. aegypti* strains exposed to TFT (Table 3 and Table 4). The mean KD_60_ of the USDA strain was 100% at LC_75_ (0.01852% or 0.06208 mg/mL), LC_99_ (0.07611% or 0.25514 mg/mL), and DC (0.15222% or 0.51028 mg/mL) and significantly higher than the KD_60_ of the Pu-Teuy strain at LC_50_ (20.8 ± 4.9%, U = 0.0, *p* = 0.003) and LC_75_ (15.0 ± 2.6%, U = 0.0. *p* = 0.002). However, the KD_60_ of the USDA strain was not significantly different from the Pu-Teuy strain at LC_99_ (80.0 ± 7.3%, U = 0.0, *p* = 0.002) and DC (89.2 ± 5.2%, U = 3.0, *p* = 0.007). Most importantly, the accuracy of HITSS-TOX was demonstrated by the mean 100% MT value for the USDA strain at LC_99_ and DC compared to LC_50_ (45.0 ± 3.7%) and LC_75_ (65.8 ± 7.0%) of the same strain. The Pu-Teuy *Ae. aegypti* had reduced susceptibility to TFT and a significantly lower MT at LC_50_ (12.5 ± 4.4%; t = 5.665, df = 10, *p* = 0.000), LC_75_ (9.2 ± 3.5%; t = 4.844, df = 10, *p* = 0.001), LC_99_ (55.0 ± 9.9%; t = 4.538, df = 5, *p* = 0.006), and DC (75.0 ± 5.2%; U = 3.0, *p* = 0.007) compared to the USDA strain (Table 4).

### 3.3. Comparison of Discriminating Concentrations of TFT and MFT Toxicity

The MT rates of Ae. aegypti (Pu-Teuy strain) were not significantly different between the DCs of TFT (0.15222% or 0.51028 mg/mL) and MFT (0.03242% or 0.10868 mg/mL) (U = 8.0, *p* = 0.087). However, the DC of 0.15222% TFT had a significantly higher KD_60_ compared to the DC of 0.03242% MFT (t = −3.102, df = 10, *p* = 0.011), as shown in Figure 2.

## 4. Discussion

This study validated the HITSS-TOX assay and provided consistent results for the responses of the *Ae. aegypti* USDA strain to TFT and MFT. First, the assay demonstrated clear dose-dependent responses for both VPs. However, each VP had different endpoints, such as KD_60_, 24 h MT, and MRE, indicating different toxicological properties. In addition, the HITSS-TOX assay accurately generated LCs and DC levels of TFT for the USDA and Pu-Teuy strains of *Ae. aegypti*. These findings indicated that the differential contact toxicity of the two chemicals reflected different ranges of SLCs, resulting in recovery at 24 h from the KD_60_ of mosquitoes exposed to lower doses of VP.

Whilst our study showed that the field-collected *Ae. aegypti* population has developed cross-resistance against permethrin and metofluthrin, this is consistent with previous findings of significantly lowered 24 h mortality against metofluthrin relative to permethrin [21]. However, mosquito recovery rate data were not presented in the CDC bottle assay [20]. Although they reported >90% mortality within an hour of exposure [20], the 24 h recovery observation should be conducted as we found that the TFT showed high knockdown with high MRE with a resistant mosquito population [33].

In a previous study, both TFT and MTF were consistently effective at low concentrations; however, MFT was approximately 10-fold less active than TFT [26]. They concluded that the differences in performance between the two AIs may have been related to different vapor pressures (TFT: 1.32 × 10^−5^ mm Hg; MFT: 3.83 × 10**^−^**^5^ mm Hg) because they confirmed a positive correlation between vapor pressure and toxicity. Furthermore, these differences may have affected the efficacy of SRs from a distance. A study conducted in an experimental hut using a 0.00625% MFT coil resulted in 58% spatial repellency against a laboratory-reared *Ae. aegypti* strain at 10 m from an outdoor site to human-occupied indoor areas [34]. The high KD_60_ (99.2%) and 24 h MT (96.7%) of 0.00853% MFT against the USDA strain could be considered as a SLC for mosquitoes not involving direct contact. The reduction in the number of female mosquitoes coming into contact with the host is defined as landing inhibition (LI) [35]. Various studies have reported that MFT reduced the landing rates of mosquitoes. For example, wind tunnel assays showed that paper strip emanators treated with 200 mg of MFT resulted in 89–91% reductions in *Ae. aegypti* landing rates [35]. In addition, polyethylene netting impregnated with 10% MFT reduced landing rates to 0–2.5% in the first 10 min of exposure in trial houses [36]. A spillover effect of MFT against *Ae. aegypti*, with high MT rates from the treated room to neighboring rooms has been reported [37]. Notably, when *Ae. aegypti* was exposed to 5% or 10% MFT in treated rooms, rapid KD_60_ and high MT (80–90%) were observed within 1 h, including behavioral impacts within minutes, such as disorientation, LI, and diversion to untreated resting sites [38].

Sublethal doses of insecticides have been shown to have potential behavioral effects, such as modifications in mating, host-finding, and feeding [39], including disarming which is defined as an adult female mosquito becoming incapacitated through either KD_60_ (reversible incapacitation due to sublethal exposure to neurotoxic compounds) or prolonged disruption of its odor receptor neurons by spatial repellents. Mosquitoes are unable to host-seek until the next night but are not killed, providing both personal and community protection and an effect on fecundity, such as a decrease in the number of viable eggs produced by a blood-fed adult female mosquito [39] or reduction of its responsiveness to oviposition attractants because of the sublethal effects of the AIs [40]. However, further study is required to provide information on the fecundity of female mosquitoes post exposure to VPs. MFT strips resulted in a rapid decrease in mosquito incidence of human biting by *Ae. aegypti* and *Culex quinquefasciatus* [41]. The quick KD_60_ and lethal effects of MFT-impregnated plastic strips on Anopheles gambiae resulted in a 98.7% reduction of the mosquito strain in a treated room along with a longer duration of repellency (up to 18 weeks) that was due to the increased MFT dosage [42]. The spatial repellency of multilayer paper strips impregnated with 200 mg of MFT was effective for over 6 weeks, resulting in >80% reduction of *Ae. albopictus* compared to 5 weeks with 200 mg TFT [15]. However, a longer residual protection efficacy of >90% for up to 6 months was observed with 10 mL of TFT-treated hessian strips against *An. arabiensis* [16]. In addition, a comparison study of TFT and MFT using topical application assay against *Cx. quinquefasciatus* and reported around a 4-fold difference in the 50% lethal doses (LD_50_) [43]. This result agreed with the current findings that TFT produced a 4.7-fold higher DC compared to MFT using the HITSS-TOX assay (Table 3). It was emphasized that sub-lethal doses of airborne TFT could produce spatial-repellent behaviors against *Ae. aegypti* [25]. Although outstanding effects were reported from these efficacy studies, the performance of MFT-impregnated net (MIN or Mushikonazu^TM^) was poor as it did not repel *Ae. aegypti* at 1 week after treatment [44] or Phlebotomine sand flies [45]. The sub-optimal concentration and mosquito resistance status may directly contribute to the inferior efficacy of MFT products. Furthermore, environmental conditions (such as temperature, humidity, and wind speed) and the target insect species may contribute to control failure [46].

Various bioassays have been introduced and widely used for susceptibility tests, including the WHO tube bioassay for generating the initial dose response for a mosquito strain with a range of concentrations or an initial baseline preliminary test performed with a full range, followed by selecting concentrations for baseline assessment [47]. Well-designed bioassays are crucial to obtain meaningful and statistically valid data that consider the number of mosquitoes per test unit, mosquito age, replicates, and concentrations for dose-response experiments to achieve maximum precision [48,49], as well as to generate more robust data [47]. The current study investigated the capacity of HITSS-TOX to evaluate contact toxicity responses of susceptible *Ae. aegypti* to a range of DCs of MFT (0.03242% (in the range 0.02624–0.04246%)) and TFT (0.15222% (in the range 0.10910–0.23942%)). However, the DC successfully established by HITSS-TOX in the current study (0.15222% or 0.51028 mg/mL) was not consistent with a previous study [17] that reported on a WHO tube bioassay that produced a 2.2-fold higher lethal concentration and DC of TFT (0.06824% or 0.22876 mg/mL) against *Ae. aegypti* USDA. These differences could be explained by the testing methods, the size and volume of the chamber, larval rearing conditions, such as overcrowding or poor diet [50,51], the time of day for testing [52], a decrease in temperature [53], an increase in humidity during insecticide susceptibility testing, the natural plant diet of the adults [13], and the genotypic background of the inbred mosquito strain. It is crucial that these physical and environmental conditions be kept constant while carrying out the HITTS-TOX bioassay to minimize multiple potential sources of variability, which can influence the result of the bioassay. As the current testing was carried out during the day which is the typical biting time by Aedes mosquitoes when they are more metabolically active, this was unlikely to be a major source of variability in the data generated using the HITTS-TOX method. The current study involved consistent monitoring of other potential factors, such as mosquito age, nutritional status, temperature and humidity, and light, as described in the experimental methods. Despite the adoption of a WHO-standard methodology, wide variations in lethal concentration distributions have been observed in a multi-center study to standardize DCs and in the development of a new requirement for a uniform coating of glass bottles [18]. For example, the report on WHO bottle bioassays conducted in six different laboratories following the same protocols (drying time, exposure time, temperature, and humidity) using nine insecticides, including TFT, resulted in different concentration MT curves between the Indian Council of Medical Research, National Institute of Malaria Research, New Delhi, India (0.70 (0.60–0.75 µg/bottle) and the Institute de Recherche pour le Development (1.5 (1.2–2.3) µg/bottle at LC_99_). This could have been due to the genotypic background of the same strains held in different laboratories displaying intraspecific variability, including adaptations to the laboratory, contamination, selection pressure during rearing conditions, genetic bottlenecks, and genetic drift. The reliability and reproducibility of data can be improved by supervising trained or experienced workers in conducting the susceptibility tests to be as consistent as possible when performing HITTS-TOX, especially where it is not possible to control the conditions fully. This is necessary at least to understand the effect that external factors can have on the outputs from this testing and to report the environmental conditions alongside the data so that the results can be interpreted accordingly [47]. Furthermore, SLCs of TFT-exposed *Ae. aegypti* female mosquitoes enhanced their oviposition behavior [54] or can associate the olfactory stimulus of pesticides with their detrimental effects and subsequently avoid pesticide contact [55]. Thus, low concentrations of chemicals not only serve to repel but also induce behavioral changes. Therefore, multi-purposed devices, such as HITSS, are necessary to understand the chemical properties and physiological aspects of post-exposed insects. Again, the flexibility of the HITSS chambers to be reconfigured for either contact irritant or spatial repellent test modality increases the utility of the tool and is an added advantage. The DCs of TFT and MFT obtained in the current study using the HITSS-TOX assay showed significantly reduced contact toxicity effects on the pyrethroid-resistant field strain compared to the pyrethroid-susceptible laboratory strain. Although the DC of TFT was 4.7-fold higher than that of MFT, the 24 h MT rates were not significantly different. However, the former resulted in a significantly higher KD_60_ than the latter. This suggested that the lethal concentration can be considered as SLCs for the pyrethroid-resistant field strain because a wider range of TFT SLCs including DC was repeatedly used for the dose-response experiments of TFT. The different volatile properties and vapor pressure between the two VPs may explain the divergent LCs and DCs.

There is a lack of specific guidance on the evaluation of products targeting *Aedes* mosquitoes, partly due to insufficient available methods to monitor *Aedes* strains, so measuring the entomological impact is difficult. Furthermore, vector control products or control efforts aimed at *Aedes* are very rarely used in isolation, necessitating the evaluation of integrated approaches, and making the link between bioassay results and predicted impact more complicated. It is a similar story for products used in larval control and for commercial products, such as emanators or spatial sprays, as well as newer classes under evaluation [56]. The current study successfully demonstrated and evaluated the accuracy of the HITSS-TOX for TFT and MFT. The optimized LCs and SLCs of each chemical using HITSS-TOX can be applied further to observe behavioral responses using HITSS-CIA and -SRA. However, both HITSS-CIA and -SRA should critically be linked with repel (respond) and KD (no-respond) concentrations. Given that different mosquito populations, including *Ae. aegypti*, respond differently to chemicals used in control strategies due to biological and environmental variations [18], the DC determined in this study might not be applicable to all mosquito populations.

## 5. Conclusions

The ability of HITSS-TOX was effectively demonstrated to perform toxicity assays for the two highly volatile pyrethroids, TFT and MFT, against both pyrethroid-susceptible and pyrethroid-resistant *Ae. aegypti*. The baseline DCs established can be used for future investigations to better understand the efficacy of various repellents against *Ae. aegypti*. The results provided optimized concentration ranges associated with KD and repellency which are crucial for conducting contact and non-contact irritancy tests using HITSS-TOX. The potential of HITSS for elucidating the behavioral responses of *Ae. aegypti* exposed to SRs will clarify the physiological status of repelled or surviving mosquitoes, including their host-seeking, blood-feeding, and ovipositing behaviors.

## Figures and Tables

**Figure 1 insects-14-00767-f001:**
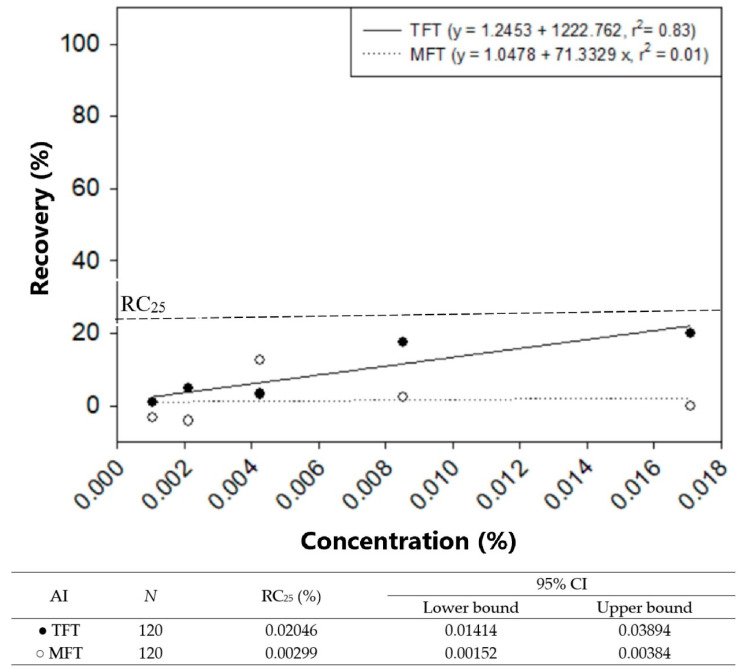
Comparison of mosquito recovery rates (MRE) of USDA *Aedes aegypti* exposed to discriminating concentrations of active ingredients (AIs) transfluthrin (TFT, closed circle), and metofluthrin (MFT, open circle) concentration (%) air-dried impregnated filter papers. R: Coefficient of determination, N: number of samples, RC_25_: 25% recovery concentration, CI: confidence interval.

**Figure 2 insects-14-00767-f002:**
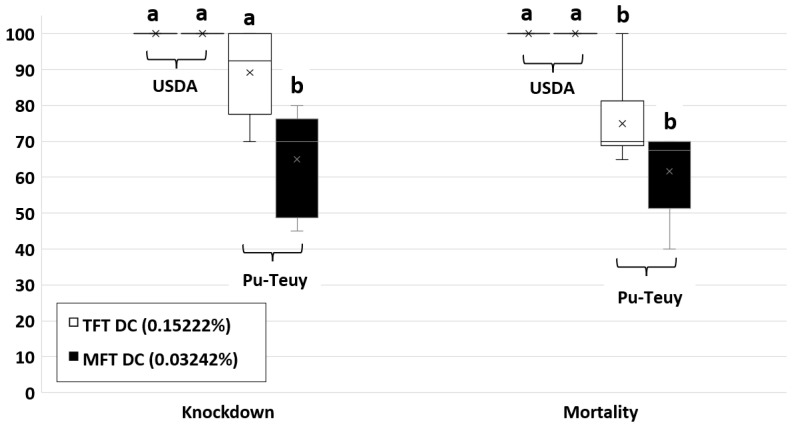
Boxplots of 1 h KD (Knockdown) and 24 h MT (Mortality) percentage of pyrethroid-susceptible (USDA) and pyrethroid-resistant (Pu-Teuy) *Aedes aegypti* exposed to discriminating concentrations of transfluthrin (TFT, 0.15222%) and metofluthrin (MFT, 0.03242%) using high-throughput screening system-toxicity assay. Different letters represent significant differences based on ANOVA (*p* < 0.05).

**Table 1 insects-14-00767-t001:** Percentage KD_60_ and 24 h MT of pyrethroid-susceptible *Aedes aegypti* (USDA) exposed to 5 serial concentrations (%) of transfluthrin (TFT) and metofluthrin (MFT) on filter paper at 1 h air-drying time using high-throughput screening system toxicity assay.

Concentration (%)	KD_60_ (%)					MT (%)				
	TFT		MFT		*p*-Value **	TFT		MFT		*p*-Value **
	Mean (SE)	Mean Rank * (Min.–Max.)	Mean (SE)	Mean Rank * (Min.–Max.)		Mean (SE)	Mean Rank * (Min.–Max.)	Mean (SE)	Mean Rank * (Min.–Max.)	
0.00107	1.7	5.0 a, A	6.7	4.9 a, A	0.120	0.8	6.2 a, A	10.0	5.7 a, A	0.152
	(1.1)	(0.0–5.0)	(2.5)	(0.0–15.0)		(0.8)	(0.0–5.0)	(6.3)	(0.0–40.0)	
0.00213	6.7	9.0 a, A	24.2	8.8 a, A	0.097	1.7	7.3 a, A	28.3	8.9 a, A	0.087
	(2.5)	(0.0–15.0)	(8.5)	(0.0–55.0)		(1.1)	(0.0–5.0)	(11.9)	(0.0–75.0)	
0.00427	17.5	14.8 ab, A	73.3	17.2 ab, B	0.006 **	14.2	15.6 ab, A	60.8	14.5 ab, B	0.020 **
	(3.6)	(10.0–30.0)	(12.6)	(20.0–100.0)		(3.5)	(5.0–25.0)	(14.2)	(15.0–95.0)	
0.00853	63.3	22.1 bc, A	99.2	23.3 b, B	0.003 **	45.8	22.6 bc, A	96.7	23.4 bc, B	0.005 **
	(8.4)	(25.0–85.0)	(0.8)	(95.0–100.0)		(12.7)	(15.0–85.0)	(2.5)	(85.0–100.0)	
0.01706	89.2	26.7 c, A	99.2	23.3 b, A	0.072	69.2	25.8 c, A	99.2	25.0 c, B	0.003 **
	(6.2)	(60.0–100.0)	(0.8)	(95.0–100.0)		(5.1)	(60.0–85.0)	(0.8)	(95.0–100.0)	

* Different lowercase letters represent significant differences among different concentrations of each chemical for KD_60_ (%) and 24 h MT (%) using Kruskal–Wallis *H* test with pairwise multiple comparison in SPSS for *p*-value < 0.05. ** Different uppercase letters represent significant differences between TFT and MFT of each concentration for KD_60_ (%) and 24 h MT (%) using Student’s *t*-test or Mann–Whitney *U* test in SPSS for *p*-value < 0.05. Analyses based on results for normality test (*p* > 0.05). SE: standard error of the mean percentage. USDA: United States Department of Agriculture. KD_60_: knockdown rate of mosquitoes exposed to TFT or MFT for 60 min. 24 h. MT: mortality rate of mosquitoes 24 h post-exposure to TFT or MFT.

**Table 2 insects-14-00767-t002:** Percentage mosquito recovery (MRE) of pyrethroid-susceptible *Aedes aegypti* (USDA) exposed to five serial concentrations (%) of transfluthrin (TFT) and metofluthrin (MFT).

Concentration (%)	MRE (%) *				*p*-Value ***
	TFT		MFT		
	Mean (SE)	Mean Rank ** (Min.–Max.)	Mean (SE)	Mean Rank ** (Min.–Max.)	
0.00107	0.8	10.17 a	−3.3	13.50 a, A	0.598
	(0.8)	(0.0–5.0)	(4.4)	(−25.0–5.0)	
0.00213	5.0	14.58 a	−4.2	12.00 a, A	0.091
	(2.9)	(−5.0–15.0)	(4.0)	(−20.0–5.0)	
0.00427	3.3	13.33 a	12.5	20.17 a, B	0.317
	(4.8)	(−10.0–20.0)	(7.3)	(−5.0–40.0)	
0.00853	17.5	18.92 a	2.5	17.83 a, B	0.212
	(8.3)	(−10.0–40.0)	(1.7)	(0.0–10.0)	
0.01706	20.0	20.50 a	0.0	14.00 a, A	0.021 *
	(7.1)	(0.0–35.0)	(0.0)	(0.0–0.0)	

* MRE% = KD% − MT% (where MT% is shown in Table 1), ** different lowercase letters represent significant differences among different concentrations of each chemical for KD_60_ (%) and 24 h MT (%) using Kruskal–Wallis *H* test with pairwise multiple comparison in SPSS for *p*-value < 0.05. *** Different uppercase letters represent significant differences between TFT and MFT of each concentration for KD_60_ (%) and 24 h MT (%) using Student’s *t*-test or Mann–Whitney *U* test in SPSS for *p*-value < 0.05. Analyses based on results for normality test (*p* > 0.05).

**Table 3 insects-14-00767-t003:** Lethal concentrations (LCs) and discriminating concentrations (DCs) of laboratory strain of *Aedes aegypti* (USDA) exposed to active ingredients (AIs).

AIs	% LC_50_ (95% FL)	% LC_75_ (95% FL)	% LC_99_ (95% FL)	%DCs	χ2(df) *	*p*-Value
TFT	0.01040 (0.00925–0.01189) a	0.01852 (0.01576–0.02280) a	0.07611 (0.05455–0.11971) a	0.15222	3.217 (3)	0.359
MFT	0.00307 (0.00279–0.00338) b	0.00497 (0.00447–0.00562) b	0.01621 (0.01312–0.02123) b	0.03242	7.692 (3)	0.053

N: number = 120 for each time point per concentration (20 per replicate). LC: lethal concentration, FL: fiducial limit; DC: discriminating concentration. * χ2: Pearson goodness-of-fit test statistic; df: degrees of freedom. Different letters indicate significant (*p* < 0.05) differences in columns based on the FL range.

**Table 4 insects-14-00767-t004:** Percentage KD_60_ and 24 h MT of USDA and Pu-Teuy *Aedes aegypti* exposed to transfluthrin-treated filter papers at different concentrations (%).

Conc. (%)	Mean (SE)					
	KD_60_ (%) *			MT (%) *		
	USDA	Pu-Teuy	*p*-Value **	USDA	Pu-Teuy	*p*-Value **
0.01040 (LC_50_)	98.3 (1.1) a, A	20.8 (4.9) a, B	0.003 **	45.0 (3.7) a, A	12.5 (4.4) a, B	0.000 **
0.01852 (LC_75_)	100.0 (0.0) b, A	15.0 (2.6) a, B	0.000 **	65.8 (7.0) b, A	9.2 (3.5) a, B	0.001 **
0.07611 (LC_99_)	100.0 (0.0) b, A	80.0 (7.3) b, A	0.041	100.0 (0.0) c, A	55.0 (9.9) b, B	0.006 **
0.15222 (DC)	100.0 (0.0) b, A	89.2 (5.2) b, A	0.093	100.0 (0.0) c, A	75.0 (5.2) b, B	0.007 **

* Different lowercase letters represent significant differences among different concentrations of each strain of *Aedes aegypti* for KD_60_ (%) and 24 h MT (%) using one-way ANOVA with Dunnett T3 or Kruskal–Wallis *H* test with multiple comparisons in SPSS when *p*-value < 0.05. Analyses based on results for normality test (*p* > 0.05). ** Different uppercase letters represent significant differences between USDA and Pu-Teuy strains of each concentration for KD_60_ (%) and 24 h MT (%) using Student’s *t*-test or Mann–Whitney *U* test in SPSS when *p*-value < 0.05. Analyses based on results for normality test (*p* > 0.05). SE: standard error of mean percentage. USDA: United States Department of Agriculture. LC: lethal concentration, DC: discriminating concentration, KD_60_: knockdown rate of mosquitoes exposed to tranfluthrin for 60 min. 24 h. MT: mortality rate of mosquitoes 24 h post exposure to transfluthrin.

## Data Availability

The datasets supporting the conclusions of this article are included within the article. Raw data are available from the corresponding author upon reasonable request.

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
