# Peer review of "Transfluthrin and Metofluthrin as Effective Repellents against Pyrethroid-Susceptible and Pyrethroid-Resistant Aedes aegypti (L.) (Diptera: Culicidae)"

_insects, 2023, doi:10.3390/insects14090767_

Round 1
Reviewer 1 Report
Kim et al. initiated this study to determine discriminating concentrations and toxicity of two pyrethroid based spatial repellents, transfluthrin and metofluthrin, using what they describe as a high throughput screening system toxicity bioassay. The authors tested several concentrations of the SR and two Aedes aegypti populations, permethrin- susceptible and resistant. It is unclear why the study was performed as there have already been studies about the toxicity of these SR and the MT dose for the specific populations tested. Addressing these studies and presenting why this study is different from what has already been published should be included in the introduction of the manuscript. Also there should be some introduction to permethrin resistance and susceptibility.
Line 60-63 this statement is confusing. The previous sentence says the test has already been shown to be advantageous with SRA. Why then is the data on SR absent? Please clarify.
Line 85-86 include the generation that was used in experiments.
Line 98 how many larvae per tray? Were the mosquitoes used in the assay the same size?
Line 114 Explain why the carrier oil is needed in addition to acetone to dilute the SR. The WHO spatial repellent assays generally use acetone alone.
Line 123 Including pictures or schematics of the device to determine how it was used would improve review of the results. Explain how the mosquitoes did not come in contact with the SR infused paper.
Line 135 Explain if replicates were done simultaneously or consecutively? Were they biological replicates or technical?
Line 162 This is not clear. Revised this sentence. Perhaps "...the lowest concentration (% or 039 mg/ml) TFT .......the highest (% or mg/ml) TFT ranged from...."
Line 164 clarify that not every concentration significantly different from each other.
Line 202 This is already mentioned in the methods section. No need to repeat it here and in all tables.
Table 2- There are no upper case letters for TFT MRE%? If not remove the commas.
Discussion It is known that different mosquito populations, including Ae aegypti, respond differently to chemicals used in control strategies. A discussion of this and how the DC determined in this study might not be applicable to all populations should be included.
The English is acceptable.
